# Clinicopathological Significances and Prognostic Role of Intratumoral Budding in Colorectal Cancers

**DOI:** 10.3390/jcm11195540

**Published:** 2022-09-21

**Authors:** Guhyun Kang, Jung-Soo Pyo, Nae Yu Kim, Dong-Wook Kang

**Affiliations:** 1Department of Pathology, Daehang Hospital, Seoul 06699, Korea; 2Department of Pathology, Uijeongbu Eulji Medical Center, Eulji University School of Medicine, Uijeongbu-si 11759, Korea; 3Department of Internal Medicine, Uijeongbu Eulji Medical Center, Eulji University School of Medicine, Uijeongbu-si 11759, Korea; 4Department of Pathology, Chungnam National University Sejong Hospital, 20 Bodeum 7-ro, Sejong 30099, Korea; 5Department of Pathology, Chungnam National University School of Medicine, 266 Munhwa Street, Daejeon 35015, Korea

**Keywords:** colorectal cancer, budding, intratumoral, prognosis, meta-analysis

## Abstract

Background: This study aims to evaluate the clinicopathological significance and prognostic implications of intratumoral budding (ITB) in colorectal cancers (CRCs) through a meta-analysis. Methods: We performed the meta-analysis using 13 eligible studies and investigated the rates of CRCs with high ITB. The correlation between ITB and clinicopathological characteristics, including disease-free survival, was evaluated. Results: The estimated rate of CRCs with high ITB was 0.233 (95% confidence interval (CI) 0.177–0.299) in overall CRCs. High ITB was significantly correlated with tumor grade, lymphatic invasion, perineural invasion, pT stage, and lymph node metastasis. In addition, ITBs were more frequently found in medullary and signet-ring cell carcinomas than in conventional adenocarcinomas and mucinous carcinomas. However, the high ITB rate was not correlated with tumor border, tumor-infiltrating lymphocytes, or microsatellite instability. CRCs with a good response after neoadjuvant therapy revealed a lower rate of high ITB than those with a poor response (hazard ratio (HR) 0.114, 95% CI 0.070–0.179 vs. 0.321, 95% CI 0.204–0.467). In addition, CRCs with high ITB had a worse disease-free survival than those with low ITB (HR 1.426, 95% CI 1.092–1.863). Conclusions: The ITB was significantly correlated with aggressive tumor behaviors and a worse prognosis in CRCs. The detection of ITB, as a histological parameter, can be useful for predicting clinicopathologic features and the prognosis of CRC.

## 1. Introduction

Tumor budding, which is the invasion of adjacent stroma by small clusters of tumor cells, is one of the critical features of malignant tumors [1]. It is defined as the spread of a single tumor cell or a small cluster of up to five tumor cells [2,3]. Tumor budding is easily found at the invasive front or within the tumor [4]. Intratumoral budding (ITB) was described by Morodomi et al. in 1989 [5]. They reported that ITB was found in 46.4% of rectal cancers and correlated with lymph node metastasis [5].

Tumor budding has been significantly correlated with aggressive tumor behaviors, including lymph node metastasis, locoregional recurrences, and poor disease-free survival [6,7]. Histological features, such as infiltrating or pushing borders, can be associated with tumor budding [8]. However, the correlation between ITB and the histological features of colorectal cancers (CRCs) is not clear. Intratumoral and peritumoral lymphocytic infiltrations may frequently be detected in CRCs [9,10]. In addition, the tumor-stroma ratio can affect the detection of ITB in CRCs. Tumor budding is evaluated in biopsied or surgical specimens. Biopsy specimens obtained from superficial areas of the tumor have limited capacity to assess peritumoral budding (PTB). In rectal cancers with preoperative neoadjuvant therapy, the clinicopathological implication of ITB rather than PTB may be important. Recently, because the detection and curation of early CRCs are increasing, the pretreatment evaluation of tumor behaviors and prognosis may be more critical [4].

Rogers et al. reported the impact of tumor budding in CRCs through a meta-analysis [11]. However, it is not clear the significance of ITB from their meta-analysis. This study aimed to evaluate the clinicopathological significance and prognostic implications of ITB in CRCs through a meta-analysis. In detail, a subgroup analysis based on evaluating criteria, study location, and tumor location was performed. The correlations between ITB and clinicopathological characteristics were investigated. In addition, the prognostic impact of ITB was elucidated for CRCs.

## 2. Materials and Methods

### 2.1. Published Study Search and Selection Criteria

Relevant articles were obtained by searching the PubMed database through 15 May 2022. We used the following keywords: “(colon or colorectal) AND (intra-tumoral or intratumoral) AND (budding or bud).” The titles and abstracts of all searched articles were screened for inclusion or exclusion. Included articles needed information on clinicopathological characteristics or prognosis in CRC with ITB. However, non-original articles, such as case reports and review articles, were excluded. In addition, those not written in English were not included in the present study.

### 2.2. Data Extraction

In the search and review, 13 eligible studies were finally included [4,12,13,14,15,16,17,18,19,20,21,22,23]. The extracted data from the eligible studies were the author’s information, publication year, study location, number of patients analyzed, the prevalence and the clinicopathological characteristics of ITB, the correlation with various markers, and the information for disease-free survival. For the quantitative aggregation of the survival results, the correlation of disease-free survival between CRC with ITBs and CRC without ITBs was analyzed according to the hazard ratio (HR) using one of two methods. In studies that did not record the HR or confidence interval (CI), we calculated these variables using the HR point estimate, the log-rank statistic or its *p*-value, and the O-E statistic (the difference between the number of observed and expected events) or its variance. If these data were unavailable, the HR was estimated using the total number of events, the number of patients at risk in each group, and the log-rank statistic or its *p*-value. The published survival curves were read independently by two authors to reduce variability. The HRs were then combined using Peto’s method [24]. All data were obtained by two independent authors (Kang G.H. and Pyo J.S.).

### 2.3. Statistical Analyses

The meta-analysis was performed using the Comprehensive Meta-Analysis software package (Biostat, Englewood, NJ, USA). The rates of CRCs with high ITB were investigated through eligible studies. Subgroup analyses based on various clinicopathological parameters were performed. The analyzed subgroups included study location, sex, age group, tumor location, tumor grading, histological subtype, lymphatic invasion, vascular invasion, perineural invasion, lymph node and distant metastasis, pTNM stages, tumor border, tumor-infiltrating lymphocytes, microsatellite instability, and tumor regression grading after neoadjuvant therapy. Heterogeneity between the studies was checked by the Q and I^2^ statistics and expressed as *p*-values. Additionally, sensitivity analysis was conducted to assess the heterogeneity of the eligible studies and the impact of each study on the combined effects. As the eligible studies used various populations, a random-effect model rather than a fixed-effect model was more suitable. Begg’s funnel plot and Egger’s test were used to assess the publication bias; if significant, the fail-safe N and trim-fill tests were additionally used to confirm the publication bias. The results were considered statistically significant at *p* < 0.05.

## 3. Results

### 3.1. Selection and Characteristics of Studies

In total, 38 relevant articles were found from the primary search using the PubMed database. In screening and reviewing, we excluded 19 articles through screening and full-text review. Of these, 11 had no information or insufficient information; 4 were excluded because they were not original, and 4 were excluded for the following reasons: articles reporting other diseases (*n* = 2) and non-human studies (*n* = 2) (Figure 1). Finally, 13 eligible articles and 4756 patients were included in the meta-analysis (Table 1). In 5 out of 13 eligible studies, ITB was evaluated based on the International Tumor Budding Consensus Conference (ITBCC) recommendation.

### 3.2. Clinicopathological and Prognostic Significance of Intratumoral Budding

The estimated rate of high ITB was 0.233 (95% CI 0.177–0.299; Table 2). In rectal cancer, the high ITB rate was 0.227 (95% CI 0.177–0.286). In subgroup analysis based on study location, the estimated rates of high ITB were 0.258 (95% CI 0.173–0.368), 0.200 (95% CI 0.152–0.258), and 0.229 (95% CI 0.152–0.331) in America, Asia, and Europe, respectively.

Next, the correlations between ITB and clinicopathological characteristics were investigated (Table 3 and Table 4). High ITB rates were 0.464 (95% CI 0.293–0.643) and 0.156 (95% CI 0.099–0.238) in CRC with high and low tumor grading, respectively. There was a significant difference between high and low tumor grading subgroups (*p* < 0.001 in the meta-regression test). High ITB was frequently found in medullary and signet-ring cell carcinomas than in conventional adenocarcinomas and mucinous carcinomas (*p* < 0.001 in the meta-regression test). High ITB was significantly correlated with high tumor grade, lymphatic invasion, perineural invasion, higher pT stage, and lymph node metastasis. However, high ITB was not correlated with tumor border, tumor-infiltrating lymphocytes, or microsatellite instability. CRCs with good responses (complete and near-complete responses) after preoperative chemoradiation therapy had lower rates of high ITB than those with poor responses (partial and poor/no responses) (0.114, 95% CI 0.070–0.179 vs. 0.321, 95% CI 0.204–0.467).

CRCs with high ITBs had a worse disease-free survival than those with low ITBs (HR 1.426, 95% CI 1.092–1.863) (Table 5). In the rectal cancer subgroup, high ITB was significantly correlated with worse disease-free survival (HR 3.412, 95% CI 1.583–7.356). In addition, in metastatic CRCs, the HR was 1.224 (95% CI 1.009–1.484). In addition, there was a significant correlation between high ITB and worse overall survival (HR 1.857, 95% CI 1.633–2.112).

## 4. Discussion

To the best of our knowledge, the present study is the first meta-analysis to evaluate the clinicopathological and prognostic significance of ITB in CRCs. The significant findings are as follows: (1) The estimated high ITB rate was 0.233 (95% CI 0.177–0.299) in CRCs. (2) High ITB was significantly correlated with aggressive tumor behaviors. (3) In rectal cancers, a high ITB rate was significantly lower in patients with a good response to neoadjuvant therapy than in those with a poor response. (4) High ITB was significantly correlated with worse disease-free survival (HR 1.426, 95% CI 1.092–1.863).

The CAP guidelines propagate a description of the presence of tumor budding [25]. However, detailed information on tumor budding is insufficient. In these guidelines, the evaluation of ITB is not clear. ITB is defined as tumor buddings identified within the tumor (tumor center). The clinicopathological and prognostic significances of high ITB must be clarified, and the criteria for evaluating ITB should be determined. However, previous studies have assessed the significance of ITB using various criteria. A meta-analysis may be appropriate to elucidate the ITBs’ significance. It is also helpful to grasp basic information on the criteria for high ITB.

In the progression of the malignant tumor, invasion into the adjacent stroma is needed. Tumor budding can be considered an important feature of epithelial-mesenchymal transition in histological examinations [1]. In “Recommendations for reporting tumor budding in colorectal cancer based on the ITBCC 2016”, the correlation between ITB and lymph node metastasis achieved a consensus [26]. However, the definitive evidence for the correlation was not described in those recommendations. In previous studies, ITB was significantly correlated with lymph node metastasis [16,17,18,19,21,23]. The present meta-analysis elucidated the significant correlations between ITB and aggressive tumor behaviors, including lymph node metastasis. However, the evaluation was not performed on the relationship between ITB and lymph node metastasis in T1 CRCs due to a lack of information in the literature. In addition, the correlations between ITB and distant metastasis and pTNM stage were different in eligible studies. In our meta-analysis, it was not found a significant correlation between ITB and distant metastasis and the pTNM stage. Subgroup analysis was not performed on the relationship between ITB and prognosis in Stage II CRC due to a lack of information in the literature. In addition, we tried to confirm the prognostic impact of ITB through the meta-analysis. In the comparison of disease-free survival, HR was 1.426 (95% CI 1.092–1.863) in overall cases. Interestingly, HR was higher in rectal cancers than in overall cases (HR 3.412 vs. 1.426).

In the ITBCC group, it is recommended that tumor budding be evaluated through the amount of budding for each microscopic field (×20, 0.785 mm^2^) [26]. In the guidelines of the College of American Pathologists (CAP), the same method is recommended for tumor budding. However, these guidelines describe PTB but not ITB. Karamitopoulou et al. reported the 10-high-power-fields scoring method using a two-tier system [27]. The authors suggested the advantages of the 10-high-power-fields scoring method as follows: (1) High interobserver agreement. (2) Method that is similar to the mitosis count of other tumors. We investigated the high ITB rate according to the criteria for high ITB. In eligible studies, the used criteria were 2, 5, and 10 per 0.785 mm^2^. The estimated high ITB rates of these subgroups were 0.205 (95% CI 0.143–0.284), 0.321 (95% CI 0.175–0.512), and 0.222 (95% CI 0.149–0.317), respectively. However, there was no statistical difference between subgroups in the meta-regression test. Interestingly, the high ITB rate was slightly higher in the 5-criteria subgroup than in the 10-criteria subgroup (0.321, 95% CI 0.175–0.512 vs. 0.222, 95% CI 0.149–0.317). The HRs of the 5- and 10-criteria subgroups were 1.428 (95% CI 1.033–1.974) and 3.350 (95% CI 1.249–8.964), respectively. Based on our results, using a high number of criteria can be more suitable for predicting the prognosis of CRC patients using ITB. However, because using a lower number of criteria also has a prognostic impact, the three-tier system can be considered for evaluating the ITB of CRCs.

Biopsied specimens are more valuable in rectal cancers with preoperative neoadjuvant therapy. In addition, in rectal cancers, evaluating ITB rather than PTB may be important. Giger et al. reported that high ITB was 17% in biopsied specimens [14]. The high ITB rate was 0.233 (95% CI 0.177–0.299) in overall cases. In rectal cancer, a high ITB rate was estimated at 0.227 (95% CI 0.177–0.286). There was no significant difference in the high ITB rate between the overall and rectal cancer subgroups. However, the prognostic impact of ITB was more prominent in the rectal cancer subgroup than in overall cases. We compared the high ITB rates between good and poor response subgroups against neoadjuvant therapy in rectal cancers. In the present meta-analysis, all of them included the results of evaluating ITB in biopsy specimens [13,15,21,23]. However, surgical specimens after neoadjuvant therapy have a limitation in interpreting the result. In CRCs with a good response, because there are fewer tumor cells after neoadjuvant treatment, low ITB can be found. Whether evaluations are pre- or post-operative may be important: if ITB is evaluated in post-operative specimens, the results might differ. Rieger et al. reported the relevance between ITB and PTB in CRC [20]. Cases with high ITB showed high PTB. However, in cases with low ITB, low PTB was found in 50% of cases. Further evaluation of the correlation between ITB and PTB will be needed. In addition, comparing the prognostic value of ITB and PTB is considered to have important significance.

This study has some limitations. First, although the subgroup analysis based on criteria for high ITB was performed, the 10-criteria subgroup was included in only one eligible study for rectal cancers. Because rectal cancers had a higher rate of high ITB and higher HR than overall cases, further studies are needed to elucidate the clinicopathological impact of the 10-criteria subgroup. Second, ITB can show intratumoral heterogeneity, but the intratumoral heterogeneity of ITBs could not be evaluated. Third, various evaluating methods, including immunohistochemistry, were used in eligible studies, but a comparison between evaluating methods could not be carried out. Fourth, the correlation between the tumor-stroma ratio and ITB could not be investigated due to insufficient information from eligible studies. Fifth, the subgroup analysis based on the sample type of metastatic foci could not be performed due to insufficient information on eligible studies. Sixth, the subgroup analysis was not performed according to the criteria for high ITB. Further evaluation for ITB evaluation criteria will be needed by analyzing the impact on the prognosis according to detailed grading.

## 5. Conclusions

In conclusion, our results showed that ITB was significantly correlated with aggressive tumor behaviors and a worse prognosis. Evaluating ITB in biopsied and surgical specimens can help predict clinicopathological characteristics and the prognosis of CRCs.

## Figures and Tables

**Figure 1 jcm-11-05540-f001:**
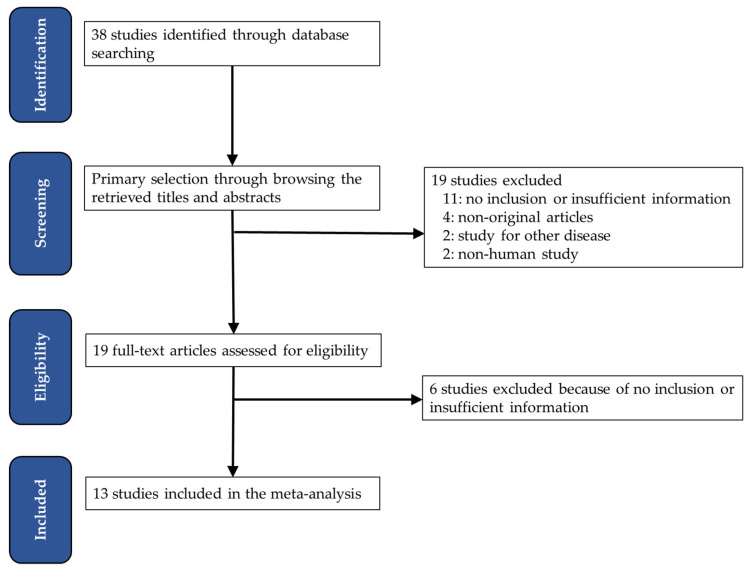
Flow chart of the searching strategy.

**Table 1 jcm-11-05540-t001:** Main characteristics of the eligible studies.

Authors	Study Location	Number of Patients	Tumor Location	Type of Sample	Criteria for High ITB
Chen 2021 [12]	USA	127	Rectum	Biopsy	2
Deb 2019 [4]	India	80	Colorectum	Resection	10
Farchoukh 2021 [13]	USA	117	Rectum	Biopsy	10
Giger 2012 [14]	Switzerland	72	Colorectum	Biopsy	ND
Huang 2019 [15]	China	141	Rectum	Biopsy	ND
Lugli 2011 [16]	Switzerland	289	Colorectum (Group 1)	Resection	≥6
		222	Colorectum (Group 2)	Resection	
Marx 2020 [17]	Germany	1262	Colorectum	Resection	10
Pour Farid 2021 [18]	Germany	245	Colorectum	Resection	10
Ramadan 2020 [19]	Turkey	122	Colorectum	Resection	10
Rieger 2017 [20]	Switzerland	215	Colorectum	Resection	ND
Rogers 2014 [21]	Ireland	89	Rectum	Biopsy	ND
Trinh 2018 [22]	Various	270	Colorectum	Resection	5
		504	mCRC (CAIRO trial)	ND	
		472	mCRC (CAIRO 2 trial)	ND	
Wen 2022 [23]	USA	529	Rectum	Biopsy	≥6

ITB, intratumoral budding; ND, no description; mCRC, metastatic colorectal cancer.

**Table 2 jcm-11-05540-t002:** The estimated rates of intratumoral budding in colorectal cancers.

	Number of Subsets	Fixed Effect (95% CI)	Heterogeneity Test (*p*-Value)	Random Effect (95% CI)	Egger’s Test (*p*-Value)	MRT (*p*-Value)
High ITB rate	12	0.222 (0.206, 0.238)	<0.001	0.233 (0.177, 0.299)	0.566	
Rectum	5	0.220 (0.186, 0.259)	0.080	0.227 (0.177, 0.286)	0.053	
Criteria, 2/0.785 mm^2^	1	0.205 (0.143, 0.284)	1.000	0.205 (0.143, 0.284)	-	0.318
Criteria, 5	3	0.344 (0.303, 0.388)	<0.001	0.321 (0.175, 0.512)	0.769	
Criteria, 10	5	0.190 (0.172, 0.209)	<0.001	0.222 (0.149, 0.317)	0.367	
America	3	0.242 (0.195, 0.297)	0.042	0.258 (0.173, 0.368)	0.068	0.742
Asia	2	0.200 (0.152, 0.258)	0.468	0.200 (0.152, 0.258)	-	
Europe	7	0.221 (0.204, 0.204)	<0.001	0.229 (0.152, 0.331)	0.715	

CI, confidence interval; MRT, meta-regression test; ITB, intratumoral budding.

**Table 3 jcm-11-05540-t003:** Clinicopathological significances of colorectal cancers with intratumoral budding.

	Number of Subsets	Fixed Effect (95% CI)	Heterogeneity Test (*p*-Value)	Random Effect (95% CI)	Egger’s Test (*p*-Value)	MRT (*p*-Value)
Sex						
Male	9	0.213 (0.192, 0.236)	<0.001	0.237 (0.155, 0.344)	0.497	0.505
Female	9	0.191 (0.171, 0.213)	<0.001	0.196 (0.133, 0.279)	0.848	
Age						
Younger	3	0.142 (0.121, 0.165)	<0.001	0.121 (0.0065, 0.214)	0.603	0.617
Older	3	0.150 (0.130, 0.173)	0.144	0.145 (0.116, 0.180)	0.469	
Tumor location						
Left-sided	8	0.168 (0.150, 0.186)	<0.001	0.171 (0.112, 0.251)	0.892	0.442
Right-sided	5	0.223 (0.197, 0.251)	<0.001	0.219 (0.132, 0.339)	0.956	
Tumor grading						
Low	7	0.146 (0.130, 0.164)	<0.001	0.156 (0.099, 0.238)	0.630	<0.001
High	6	0.457 (0.397, 0.518)	<0.001	0.464 (0.293, 0.643)	0.965	
Histologic subtype						
Conventional	2	0.137 (0.108, 0.173)	0.695	0.137 (0.108, 0.173)	-	<0.001
Mucinous	3	0.377 (0.244, 0.531)	0.586	0.377 (0.244, 0.531)	0.713	
Medullary	1	0.800 (0.459, 0.950)	1.000	0.800 (0.459, 0.950)	-	
Signet ring cell	1	0.818 (0.493, 0.954)	1.000	0.818 (0.493, 0.954)	-	
Lymphatic invasion						
Presence	3	0.245 (0.190, 0.309)	0.350	0.244 (0.188, 0.310)	0.008	<0.001
Absence	3	0.101 (0.072, 0.139)	0.798	0.101 (0.068, 0.147)	0.798	
Vascular invasion						
Presence	4	0.395 (0.317, 0.479)	<0.001	0.302 (0.137, 0.543)	0..233	0.320
Absence	4	0.231 (0.198, 0.267)	<0.001	0.190 (0.094, 0.346)	0.460	
Perineural invasion						
Presence	2	0.342 (0.186, 0.540)	0.282	0.339 (0.175, 0.554)	-	0.006
Absence	2	0.131 (0.096, 0.176)	0.577	0.131 (0.096, 0.176)	-	
pT stage						
pT1–T2	8	0.097 (0.075, 0.124)	0.065	0.100 (0.067, 0.148)	0.928	0.016
pT3–T4	8	0.222 (0.204, 0.240)	<0.001	0.227 (0.148, 0.332)	0.826	
LN metastasis						
Presence	9	0.310 (0.283, 0.338)	<0.001	0.329 (0.242, 0.430)	0.554	0.004
Absence	9	0.144 (0.124, 0.166)	<0.001	0.146 (0.086, 0.237)	0.855	
Distant metastasis						
Presence	4	0.381 (0.281, 0.492)	0.054	0.361 (0.212, 0.543)	0.352	0.067
Absence	4	0.189 (0.157, 0.226)	<0.001	0.169 (0.088, 0.299)	0.715	
pTNM stage						
Stage I–II	2	0.191 (0.137, 0.259)	<0.001	0.168 (0.031, 0.560)	-	0.288
Stage III–IV	2	0.359 (0.285, 0.442)	0.006	0.371 (0.187, 0.604)	-	

CI, confidence interval; MRT, meta-regression test; LN, lymph node.

**Table 4 jcm-11-05540-t004:** The estimated rates of intratumor budding in colorectal cancers with various conditions.

	Number of Subsets	Fixed Effect (95% CI)	Heterogeneity Test (*p*-Value)	Random Effect (95% CI)	Egger’s Test (*p*-Value)	MRT (*p*-Value)
Tumor border						
Pushing	5	0.151 (0.119, 0.190)	0.033	0.139 (0.087, 0.216)	0.423	0.208
Infiltrating	5	0.256 (0.228, 0.287)	<0.001	0.234 (0.113, 0.424)	0.824	
TIL						
High	4	0.263 (0.218, 0.314)	<0.001	0.212 (0.088, 0.430)	0.185	0.759
Low	4	0.241 (0.214, 0.270)	<0.001	0.250 (0.121, 0.446)	0.961	
MMR						
MMR deficient	5	0.252 (0.209, 0.300)	0.006	0.218 (0.140, 0.325)	0.332	0.796
MMR proficient	5	0.203 (0.181, 0.228)	<0.001	0.186 (0.094, 0.336)	0.724	
TRG						
TRG 0–1	4	0.114 (0.070, 0.179)	0.765	0.114 (0.070, 0.179)	0.039	0.004
TRG 2–3	4	0.290 (0.234, 0.352)	0.005	0.321 (0.204, 0.467)	0.023	

CI, confidence interval; MRT, meta-regression test; TIL, tumor-infiltrating lymphocyte; MMR, mismatch repair; TRG, tumor regression grade.

**Table 5 jcm-11-05540-t005:** The prognostic implications of intratumoral budding in colorectal cancers.

	Number of Subsets	Fixed Effect (95% CI)	Heterogeneity Test (*p*-Value)	Random Effect (95% CI)	Egger’s Test (*p*-Value)
Disease-free survival	6	1.043 (1.012, 1.074)	<0.001	1.426 (1.092, 1.863)	0.015
Rectal	2	3.412 (1.583, 7.356)	0.954	3.412 (1.583, 7.356)	-
Metastatic	2	1.229 (1.052, 1.435)	0.216	1.224 (1.009, 1.484)	-
Overall survival	2	1.857 (1.633, 2.112)	0.875	1.857 (1.633, 2.112)	-

CI, Confidence interval.

## Data Availability

Data are available in a publicly accessible repository.

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
