# Peer review of "Clinicopathological Significances and Prognostic Role of Intratumoral Budding in Colorectal Cancers"

_jcm, 2022, doi:10.3390/jcm11195540_

Round 1
Reviewer 1 Report
Kang et al. report a meta-analysis investigating clinicopathological significances and prognostic role of intratumoral budding in colorectal cancers. While they state that this is the first meta-analysis to evaluate the clinicopathological and prognostic significance of ITB in CRCs, meta-analyses have been published on the subject. (e.g. Rogers et al PMID: 27599041)While there are differences in the methodology, this should be clarified in the manuscript.
To improve the clinical value of the manuscript, several issues need to be addressed as below:
· With the heterogeneity in the reporting system, the manuscript should describe how each manuscript analyzed and reported ITB. A subgroup analysis of those manuscripts would also be informative. In particular, they should clarify how many manuscripts reported ITB in accordance with the International Tumor Budding Consensus Conference recommendations.
· Overall survival should be reported in addition to the DFS.
· While several studies report a correlation between tumor budding with TNM stage and distant metastasis, the meta-analysis failed to demonstrate this.
· The type of samples that were analyzed should be reported. Are they all surgical samples? In the case of metastatic disease, are they exclusively primary tumors? Was there any assessment of metastatic lesions? Was there any report of nonresectable metastatic disease? If not, this should be clarified that the metastatic disease is only limited to resectable tumors
· Some previous papers have reported the significance of tumor budding in endoscopically resected T1 tumors and their association with LN involvement. This is an important clinical finding and would be valuable to explore in this meta-analysis
· Some previous papers have reported the significance of tumor budding and its association with survival in Stage II tumors. This is important since the information can provide a valuable tool for guiding the necessity of adjuvant systemic treatment in this subset of patients, where there is clearly a need for a predictive clinical tool. This study can further look into this area.
Author Response
Dear Editor-in-Chief,
We thank the reviewers for their constructive comments on our manuscript (Manuscript ID: jcm-1873132)
We tried to address the points raised by the reviewers as best as we could. The specific responses to the reviewers’ comments are described in Reply to Reviewers. We also fixed other unintended errors in our manuscript.
Once again, we are thankful to have a chance to improve our manuscript and hope that it is good enough to be published in the Journal of Clinical Medicine.
Please see the attached file and let us know if there are any more things we need to change.
Thank you again.

Reviewer 2 Report
I commend the authors for the meta-analysis. Have any of the studies they analyzed compared the correlation between ITB and peritumortal budding? Do the authors believe that ITB is more significant or has a better prognostic value that peritumoral budding? It would be preferable if they can elaborate and comment on this in the manuscript. There is not a consensus of what is regarded as significant ITB or its grade. Thus, the analyzed articles may have different interpretations as to how they define ITB. Can the authors comment on this as well?
Author Response

(The authors gave the same response as above.)
